# India and Refugee Law: Gauging India's Position on Afghan Refugees

Atul Alexander * and Nakul Singh

Department of Law, West Bengal National University of Juridical Sciences, Kolkata 700098, West Bengal, India; nakul219106@nujs.edu
* Correspondence: atulalexander100@nujs.edu

**Abstract:** The turbulent transition of power from the Ghani administration to the Taliban regime has not only signalled a death knell to the fundamentals of representative democracy, but it has also provided fertile ground for the large-scale exodus of refugees into neighbouring nations. In view of this, a scrutiny of the Indian state's response to the influx of Afghan refugees is warranted. India is not a signatory to the 1951 Refugee Convention, nor to the 1967 Protocol, and, in the absence of any concrete national refugee law and policy, Afghans who are seeking refugee status are processed on a haphazard case-by-case basis. In chalking out a future course of action, this paper aims to analyse India's response to the possible Afghan refugee inflow in the aftermath of the Taliban takeover and in light of India's recent endorsement of the Global Compact on Refugees (GCR). Against the backdrop of the limited mandate of the UNHCR and the lack of "political will" from the successive governments, we contend that the Supreme and High Courts of India have been instrumental in construing a tentative shield of protection for persons already in the country, which is working out of a judicial form of the endorsement of the *non-refoulment principle*, in the absence of legislative and executive commitments, and the preferential "acts of kindness" strategy, which discriminates amongst different refugee groups as per origin or religious belief. Moreover, it is argued that the GCR has made few inroads into the overall paradigm as to how refugees are perceived in India. The research concludes that India must enact legislation on refugees for any constructive engagement beyond archaic quick-fix solutions.

**Keywords:** Afghanistan; India; refugees; non-refoulment; refugee convention; UNHCR

## 1. Introduction

The tumultuous transition of power from the Ghani government to the Taliban regime has brought home an unfathomable plight to the Afghans (Sajjad 2021), especially women, who will be at the receiving end of the extreme and draconian interpretation of the Sharia Law (Barr 2021), and the Shia Hazarite Muslims (Barr 2021) who will face persecution on religious grounds, as well as the individuals who worked under the aegis of and supported the military establishments, activities, plans, and operations of the United States and the NATO allies. The fear of persecution on all the dramatis personae is propelled by religious, ethnic, and political proclivities, as well as gender. Because of this well-founded fear of persecution on a wide array of grounds, a substantial number of Afghans are fleeing Afghanistan. Migration to "neighbouring" Asian countries is a logical consequence of the West failing to amicably cultivate an accommodating position towards the migration that has been fueled by the current crisis and the large movements driven by the 2015 Syrian war. The Taliban capture of Kabul has created a situation wherein there is a high likelihood of Afghan refugees flowing into India.

The influx of Afghan refugees into India has also been an occurrence in the past (i.e., during the Soviet invasion of Afghanistan). In this context, this paper undertakes futuristic research on the treatment of Afghan refugees that are flowing from Afghanistan into India; in doing so, the paper also provides a comprehensive study on India's refugee policy.

## 2. Afghanistan: A History Rife with Conflict and War

The nation of Afghanistan has had a long history of war and internal conflict, the causative factors of which have been foreign occupiers, invaders, and internal warring factions. The first to invade and conquer the land was Darius of Babylonia in the year 500 B.C., who was followed by Alexander the Great of Macedonia in 329 B.C. What followed was the violent conquest of the land by Mahmud Ghazni, who was an eleventh-century conqueror (Anglo-Afghan Wars British-Afghani History 2021). A series of conflicts began during the 19th century when Britain wanted to annex Afghanistan in order to protect its Indian empire from Russia, which resulted in the British Afghan wars in 1838–1842, 1878–1880, and 1919–1921 (Anglo-Afghan Wars British-Afghani History 2021).

Post the first world war, the British were already drained of resources and were compromised, and they had negotiated a peace treaty with the then Amir of Afghanistan, which resulted in the recognition of Afghan independence by the British (Imy 2019). Consequently, in the aftermath of the 1919–1921 battle, Afghanistan became an independent nation (Anglo-Afghan Wars British-Afghani History 2021). To ensure that Afghanistan caught up with the rest of the world, Amanullah Khan declared himself the monarch of Afghanistan and initiated a regime of wide-sweeping reforms (Amānullāh Khan Ruler of Afghanistan 2021). Unsatisfied and discontented with many of his modernisation policies, local religious sects took up arms against his administration, which led to another onslaught of violence in the country (Shahrani 2005). In the year, 1973, a military coup was orchestrated, and Khan overthrew the monarch. Khan's party, the People's Democratic Party, assumed power, and he was named president. In this setting, the Republic of Afghanistan was established with good relations with the Soviet Union.

In 1978, however, Khan was killed in a communist coup, and the Afghan communist party took charge of the nation, backed by the Soviet Union. The communist government found very little favour among the masses, and it managed to be a repository of their faith and support by instilling fear through the infliction of casualties (Ruiz 2001). To escape this "political persecution", Afghans, in the thousands, started emigrating to the nearby lands of Pakistan and Iran (Noor 2021). What also followed was the beginning of a feeling of dissatisfaction among the local conservative religious sects with the introduced reforms and, thus, they formed a guerrilla movement or, the "Mujahideens", which was assembled and trained to topple the soviet-backed government (Latynski and Wimbush 1998).

Taking stock of the fragile spot that the communist government in Afghanistan was in, and pursuant to the Brezhnev doctrine, the Soviet Union invaded Afghanistan on 24 December 1979 (Soviet Invasion of Afghanistan 1979). In light of this invasion and the consequential battle between the US-backed Mujahideens and the Soviet Union, to escape collateral damage, the mass exodus of Afghan nationals began, and the Afghans started fleeing to neighbouring countries and other parts of the world (Safri 2011).

In the year, 1992, the Soviet Union withdrew, and, in the year, 1995, (Rubinstein 1988) the Taliban, on the promises of ensuring peace, security, and stability in the region, gathered the support of the public and rose to power. Once back in power, the Taliban started persecuting the Shia minorities and imposed draconian interpretations of the Sharia Law on women. Fearing these changes, many Afghan nationals, again, decided to emigrate out of Afghanistan (Motlagh 2021).

Post 2001, America and other NATO allies invaded Afghanistan in order to battle terrorism and to weed out the Mujahideen groups that were supporting and sheltering Bin Laden, including the Taliban (The U.S. War in Afghanistan 2021). However, again, to save themselves from the collateral damage, Afghan nationals fled Afghanistan.

Afghanistan is experiencing an exodus along similar lines amidst the tumultuous transition from the Ghani government to the Taliban regime. The change has brought home an unfathomable plight to the Afghans (Sajjad 2021). especially women who, again, will be at the receiving end of the extreme and draconian interpretation of the Sharia Law (Barr 2021). and the Shia Hazarite Muslims (Hazara Shias Flee Afghanistan Fearing Taliban Persecution 2021). who will face persecution on religious grounds, as well as the

individuals who worked under the aegis of and supported the military establishments, activities, plans, and operations of the United States and the NATO allies. Because of this well-founded fear of persecution on a wide array of grounds, Afghans are fleeing Afghanistan in substantial numbers.

Nations across the globe have opened their borders to accept the inflow of Afghans who are desperately fleeing their country. A few of those nations are the United States, Germany, Italy, France, the United Kingdom, Pakistan, Iran, and India, among many others (Anglo-Afghan Wars British-Afghani History 2021). However, it is pertinent to note that European nations, after having faced the refugee crisis in 2015, have not been as amenable to the influx of refugees as they were in the 1980s and the 1990s. On taking a total stock of all the Afghan refugees worldwide from the 1980s to the 2020s, the maximum number of refugees have been accommodated by Pakistan, followed by Iran. India and the United States stand at the 12th and 22nd positions, respectively (Refugee Data Finder 2021).

### 3. India and Refugee Protection

*A Glance at the Situation and Protection of Refugees in India*

At the very outset, it must be noted that India is neither a signatory to the 1951 Refugee Convention, nor to the 1967 Protocol, despite unabating attempts on the part of the UNHCR to request India to sign the instruments (Janmyr 2021). What complicates matters further is that, in addition to India not being a signatory to either of the two instruments, not only does it not have domestic legislation to deal with the granting of refugee status to forced migrants, the term, "refugee", has not been defined in any domestic legislation of the land. However, it is pertinent to note that, even if India is not a party to any of the international instruments, apropos the granting of refugee protection and asylum, it has actively contributed to the development and endorsement of the Global Compact on Refugees (GCR), which the United Nations officially endorsed on the 17 December 2018. The affirmation of the GCR by India in light of the legislative vacuum demonstrates her willingness to usher in a uniform, fairer, and stronger course of action and procedure to accommodate and deal with large refugee movements and, consequently, to respect the principle of "burden sharing" (Xavier and Rai 2020).

Given these shortcomings, in granting refugee status to forced migrants, or in preventing their forceful deportation or refoulement, the judiciary, the executive, and the legislature have employed different—and at times contradictory—courses of action, which will be elaborated in the following sections.

Coming to the specifics of the nationalities of the refugees in India, as per the available statistics (UNHCR Factsheet 2021). on the numbers and nationalities of all the refugees that are spread across India, it is evinced that nationals of Myanmar, Sri Lanka, Tibet, Pakistan, and Afghanistan have been accommodated in refugee camps spread across the country (UNHCR Factsheet 2021). As far as the Afghan refugees are concerned, their migration in order to flee war and persecution had been in the offing right from the 1970s to the 1990s (Refugee Data Finder 2021). On the very specifics of the numbers, in the period of the Soviet invasion, from the 1970s to the 1990s, India experienced an influx of about 60,000 Afghan refugees. However, as per the latest data, as of 31st March 2021, according to the UNHCR, the number of Afghan refugees in India stands at 15,217 (UNHCR 2021).

Moreover, in the refugee camps that are spread across the country, not all of the forced migrants have been accorded the status of a refugee. Granting a forced migrant the status of a refugee, or recognising a forced migrant as a refugee, becomes relevant because, by being labelled by the executive wing of the government, an LTV (long-term visa) is granted to the refugee, which enables the person to take up private employment or to enroll in an educational institution (Refugee Data Finder 2021). Procedurally, it is the executive wing of the government (FRRO Notification 2017). that is responsible for the granting or certifying of the refugee status. In the interest of clarity, it is pertinent to note that, in the current scheme of things, e-visas, or emergency visas, are being granted rapidly in order to enable the entry of Afghan refugees into India (India Launches New Category of Online

Visa for Afghanistan 2021). A humanitarian-motivated e-visa, in and of itself, is not an LTV that has the same benefits that usually accompany it. Once the refugee is in India, in order to join an educational institution or take up private employment, an application for an LTV is to be made to the Foreigners Registration Office (FRO), and, on approval by the Ministry of Home Affairs (MHA), the LTV will be granted to the Afghan refugee (FRRO Notification 2017).

## 4. Attitudes towards Refugees

### 4.1. The Indian Executive Branch of Government

Prior to delving into the attitudes of the executive, it is to be noted that India does not have a fixed, concrete, and uniform policy vis-à-vis refugees and their concerns purporting to the determination of their refugee status, nor does it have clarity on the concept of *non-refoulement* and its availability as a right or the amount of financial assistance that is required by a refugee (Samaddar 2003). Instead, the executive has given itself a wide set of powers for deciding to whom to give protection or favourable treatment, or to whom to allow entry, as well as for deciding who are not to be allowed, or who should be deported (Jalais 2005).

Additionally, the decisions by the Indian executive on whether a group of asylum seekers that belong to a particular nationality should be granted the status of refugees or not, and whether, therefore, they should be accorded the corresponding rights and privileges, are premised on national proclivities in the fields of international politics or relations. On the basis of this, the granting of preferential treatment to one sect, and the neglect of the others, is what is commonly called, "calculated kindness", or "strategic ambiguity" (Chimni 2003).

As far as the recognition of forced migrants as refugees is concerned, Tibetans have been the largest group among the beneficiaries of the refugee certification procedure that is followed by the executive. The Afghan sect continues to remain the smallest. Even among the Afghan community, members from the Sikh and Hindu communities have been integrated into the Indian society at a much more rapid pace than others in the Afghan community (Noor 2021). For such anomalies, the executive has provided no explanations or justifications.

In terms of the responses by the Indian executive, the Indian government advanced that, in the absence of any municipal legislation to deal with refugees, the Ministry of Home Affairs (MHA), in 2011, issue a standard operating procedure. As per this procedure, the foreigner who claims to be a refugee on the well-founded fear of persecution on the grounds of race, religion, sex, ethnicity, nationality, etc., can be recommended by the state government or by the union territory administrations to the MHA for the granting of an LTV[1]. Once an LTV has been granted, the refugee is entitled to employment Lok Sabha, or to join any educational institution,[2] or can even take up Indian Citizenship under the Indian Citizenship Act, 1955.

With regard to the current Afghan refugee crisis, only the executive has stepped in and made provisions for addressing the plights and concerns of Afghan nationals who are fleeing Afghanistan. In the aftermath of the Taliban takeover, the Ministry of External Affairs (MEA) rolled out the scheme of granting emergency visas under the category of e-visas, or electronic visas, to Afghan nationals. The validity of the e-visa is for a mere six months (Ministry of Home Affairs 2021). However, the granting or sanctioning of the humanitarian-motivated e-visa does not axiomatically extend to the beneficiaries of the e-visas those benefits which the LTV visa grants to refugees. On the specifics of the humanitarian-motivated e-visa, or electronic visa scheme, this initiative was rolled out in the year, 2014, along the lines of the Indian government's flagship movement: "digital India". The humanitarian-motivated e-visa scheme makes it very convenient for interested

---

1   Lok Sabha, Un-starred Question No. 739, Answered on 15 July 2014.
2   Lok Sabha, Un-starred Question No. 7538, Answered on 22 May 2012.

travellers to apply for the visa online and to spare a visit to the embassies. The move was aimed at easing the access to the country of foreign nationals in order to promote tourism and investment. Until 2021, humanitarian-motivated e-visas were only granted for tourism, business, conferencing, and medical purposes (Indian Visa Online 2021). however, post 2021, in the aftermath of the Taliban takeover, a new category under the e-visas, called the "e-emergency visa", was introduced solely on "humanitarian grounds" in order to enable the Afghans to rapidly escape (India Issued 200 e-Emergency Visas to Afghan Nationals 2021). the torment of the Taliban and seek refuge in India.

The forced migrants will have to apply for the same visa and forward it to the state government in order to claim such benefits. Only after evaluating the application on the basis of the MHA's standard operating procedure will the applicant be accorded the benefits and the long-term visa (LTV) that is granted to the forced migrants.

In this context of discussing the specific concerns of Afghan refugees, the role of the UNHCR in India assumes prime importance. The organisation often aids the Indian executive in verifying the relevant documents of the forced migrants in order to better equip the government with the ability to grant refugee status to forced migrants. It is to be noted that even the UNHCR itself holds the prerogative of suo moto labelling the forced migrants as refugees and providing them with financial and legal assistance. The mandate of the UNHCR extends to the refugees who are not nationals of Sri Lanka, Tibet, Pakistan, and Bangladesh. Most of the refugees who are recognised and registered with the UNHCR happen to be Afghan refugees (Chimni 2003). The UNHCR is scaling up its capacity in terms of registration and assistance in order to accommodate the Afghan refugees in India. However, this mandate of the UNHCR vis-à-vis India is only to the extent of humanitarian operations, and it has not been granted formal status by the Government of India (A Pocket Guide to Refugees 2008).

Global Compact on Refugees

India voted in favour of the GCR on 17 December 2018; however, it is equally important to state that the GCR is a non-binding instrument and it came in the aftermath of the states closing their borders. Moreover, the GCR adopts a multistakeholder approach that is focused on a charity-based approach, rather than on a rights-based one. Given that the GCR has an inherent contradiction, it is unclear whether it is a political or a non-political instrument (Field and Burra 2020). As Professor Srinivas Burra opines, "...it (GCR) asserts that it is entirely non-political in nature, including in its implementation, and is in line with the purposes and principles of the Charter of the United Nations. The non-political nature of the GCR is contradicted when the Compact later states that it represents the political will and ambition of the international community as a whole for strengthened cooperation and solidarity with refugees and affected host countries" (Field and Burra 2020). Although, on a positive note, the GCR views the private investment as a key factor to the integration of the refugee into the mainstream society, it presents a challenge in that the laws in India require it to be moulded accordingly. In the context of India, the GCR does not provide any insight into the access to protection; this is one of the most critical priorities.

Moreover, the involvement of multistakeholders in the GCR could mean and imply that humanitarian organisations work alongside a state (India) to safeguard the refugees; this can create an apprehension of discrimination with regard to various social factors, as no humanitarian organisation operates on the principle of "neutrality". Therefore, it is clear that, despite the high ideal that the GCR preaches, it lacks teeth on multiple fronts, and it is dependent on the mercy of the states at the end of the day. Despite these shortcomings, the GCR certainly discourages the preferential treatment of refugees from one country over another; hence, India's strategy of calculated kindness towards some nationals over others could not stand the test of time (Global Compact on Refugees 2018). Moreover, India, being an Executive Member of the UNHCR, has an inherent moral obligation to India to safeguard the refugees.

### 4.2. *The Indian Legislature*

Eminent scholar and jurist par excellence, Professor B.S. Chimni, rightly pontificates that the influx of refugees into India is not an occurrence of the current times, but has been in place from time immemorial (Chimni 2000). The first in the series of influxes was in the year 1959, when the Dalai Lama and his fellow followers fled Tibet to avoid political and religious persecution as China began to propagate its influence on the region that we know as "Tibet" (Kaufman 2009). However, during this period, it was also evident that, as rightly put by jurist, Upendra Baxi, " . . . the human rights plights of immigrants and refugees continue to tell chilling stories about states' lethal sovereignties." (Baxi 2016).

The second in the series was experienced during the Indo-Pakistan war of 1965, when the minority communities in East Pakistan fled the nation in fear of facing persecution by the Pakistan military. It is also pertinent to note that, during this period, and specifically from 1964 to 1968, the Chakmas, in large numbers, migrated to India from the Chittagong hills (Das Gupta 1986, p. 1665). The third and the largest wave of refugee influxes was during the year, 1971. During the Bangladesh liberation war, India experienced a massive influx of nationals from erstwhile East Pakistan, coupled with the influx of the Chakmas, again, from the Chittagong hill tracts (Das Gupta 1986).

India is neither a party to the 1951 Refugee Convention, nor to the 1967 Protocol, despite such repetitive influxes. To shed light on the reasons for this would be to capture the following points: During the deluge of incoming refugees in the year of 1971, India was not provided with any aid or commitment from the Western world and was left to fend for herself. This reason reads in conjunction with the fact that India views the Refugee Convention and the Protocol as Eurocentric or Western-oriented, as it does not showcase any sensitivity towards the concerns of these developing countries on the point of the refugee influx[3], and it has only bolstered its resolve to hold its ground and to not sign the Convention. In addition to the named concern, the relevant others are national security considerations and burgeoning populations, and the problems and issues that are involved with the access to limited and constrained resources, and the changes or disturbances in the current schemes of demographics throughout the northeastern region (De Sarkar 2015). Although it is not a signatory to the major refugee Conventions, India has, considering its fixed capabilities, always tried its best to accommodate refugees and to address their concerns, and the Indian government has extensively carried this out (Noor 2021).

The document, which is the constitution of India,[4] embodies a wide array of fundamental rights in Part III that are available to all persons, irrespective of nationality, which logically extrapolates to them being applicable to refugees as well. Such fundamental rights include the right to be guaranteed equality before the law[5]; the right of protection with respect to conviction for offences[6]; the right to life and personal liberty[7]; the right of protection from arrest and detention in certain cases[8]; and the right to freedom of religion[9]

The most vital article in addressing the plight of the refugees, according to Professor Chimni, is Article 21 of the Constitution of India. According to the learned scholar, "it can be argued that Article 21 encompasses the principle of *non-refoulment* which requires that the State shall not expel or return a refugee in any manner whatsoever to the frontiers of territories where his life or freedom would be threatened on account of his race, religion, nationality, membership of a particular "social group or political opinion" (Chimni 1994).

Apart from the constitution, on the specifics of statutory laws and legislation, there is no specific legislation that defines the term "refuge", or that deals with the concerns of the refugees. However, their entries, stays, and exits are regulated by the legislation that

---

3　Lok Sabha, Un-starred Question No. 3693, Answered on 13 December 2000.
4　The Constitution of India, 1950.
5　Article 14, The Constitution of India.
6　Article 20, The Constitution of India, 1950, Art.20.
7　Article 2, The Constitution of India, 1950, Art.21.
8　Article 22, The Constitution of India, 1950, Art.22.
9　Article 25–28, The Constitution of India, 1950, Art.25–28.

pertains to foreigners in India. Chronologically, the very first one to be in place was Section 3(1) of the Passport (Entry into India) Act (1920)[10], on the basis of which any foreigner not possessing a valid passport can be imprisoned or fined, or both, vide Section 3(3). The said act also empowers the Central Government to make orders to have any person removed from the territory of India for the violation of any of the sections or rules under the said act.

Additionally, according to the Registration of Foreigners Act (1939)[11], and rules that were restructured in 1992, any foreigner (which is defined as a person who is not a citizen of India vide Section 2(a) of the said act) entering and staying in India for more than 180 days should mandatorily register themselves with the Foreigners Registration Office, and the failure to do so could result in imprisonment or a fine, or both, according to Section 5 of the said act. It is also to be noted that another legislation that is employed when dealing with refugees is the Foreigners Act (1946)[12]. As per this act, the Central Government has been empowered, under Section 3, to make rules and to order the prohibition, restriction, and regulation of the entry, stay, or departure of any foreigner or class of foreigners in India. In pursuit of the same section, the government enacted the Foreign Order (1948).[13] Clauses 3(2)(a), 4(a), 5, 7, 8, 9, 10, and 11 of the code form an entire code in themselves that deals with foreigners. These clauses, in toto, regulate the entry, stay, travel, access to public places or prohibited places, restrictions on movement and employment, and other restrictions that may be imposed in the public interest.

Apart from the existing framework of laws, the response of the Indian Parliament has constantly been that there is no compelling need for a novel specific refugee law to be enacted, and that the existing framework of laws is adequate to deal with the concerns of refugees.[14] In light of this contention, in the 14th Lok Sabha session (the lower house of our bicameral legislature), the recommendations of the National Human Rights Commission in calling for a specific framework to deal with the concerns of refugees was expressly rejected.[15]

However, in 2015, three Bills were introduced in the Lok Sabha to deal with and address the concerns of the refugees, and they were: the Asylum Bill (2015),[16] by Dr. Shashi Tharoor (Member of Parliament); the National Asylum Bill (2015),[17] by Feroz Varun Gandhi (Member of Parliament); and the Protection of Refugees and Asylum Seekers Bill (2015),[18] by Rabindra Kumar Jena (Member of Parliament).

The Asylum Bill of 2015 is introduced by Dr. Tharoor in its preamble, who pontificates that the aim and the object of the said bill are to consolidate, streamline, and harmonise the various standards and procedures that are applicable to refugees and asylum seekers in the territory of India.[19] The bill defines who a refugee is under Section 4 and Section 30, and it lays down the two determining criteria under Sections 4(a) and 4(b). Once the status of a refugee is accorded to the asylum seeker, non-refoulement, as a matter of right, is available

10    The Passport (Entry into India) Act (1920). Available at https://www.mha.gov.in/sites/default/files/PptEntryAct1920_0.pdf (accessed on 1 April 2022).

11    The Registration of Foreigners Act (1939). Available at https://www.mha.gov.in/sites/default/files/The_Registration_of_Foreigners_Act_1939.pdf (accessed on 29 March 2022).

12    The Foreigners Act (1946). Available at https://legislative.gov.in/sites/default/files/A1946-31.pdf (accessed on 29 March 2022).

13    The Foreign Order (1948). Available at https://upload.indiacode.nic.in/showfile?actid=AC_CEN_5_23_00048_194631_1523947455673&type=order&filename=The%20Foreigners%20Order%201948.pdf (accessed on 29 March 2022).

14    Rajya Sabha, Starred Question No. 2533, Answered on 16 August 2000.

15    Lok Sabha, Un-starred Question No. 277, Answered on 21 February 2006.

16    The Asylum Bill, No. 334 of 2015 (India), Introduced in Lok Sabha by Dr. Shashi Tharoor, MP.

17    The National Asylum Bill, No. 341 of 2015 (India), Introduced in Lok Sabha by Feroze Varun Gandhi, MP.

18    The Protection of Refugees and Asylum Seekers Bill, No. 290 of 2015 (India), Introduced in Lok Sabha by Rabindra Kumar Jena, MP, Available at http://164.100.47.4/billstexts/lsbilltexts/asintroduced/3024ls.pdf (accessed on 10 November 2021).

19    The Protection of Refugees and Asylum Seekers Bill, No. 290 of 2015 (India), Introduced in Lok Sabha by Rabindra Kumar Jena, MP, Available at http://164.100.47.4/billstexts/lsbilltexts/asintroduced/3024ls.pdf (accessed on 10 November 2021) at Preamble.

to that person or refugee.[20] Additionally, the bill makes available to asylum seekers, as well as refugees, inter alia access to employment[21], healthcare[22], education[23], and freedom from discrimination.[24]

On the other hand, The National Asylum Bill of 2015 endeavours solely to provide rules and regulations for granting citizenship to refugees and asylum seekers.[25] The definition of, "refugee", in the said bill is akin to the definition that is employed in the 1951 Refugee Convention, but the catch is that it excludes any person who does not possess a nationality.[26]

Meanwhile, the Protection of Refugees and Asylum Seekers Bill (2015) endeavours to establish an appropriate and uniform legal framework to deal with matters that pertain to forced migration, the determination of the refugee status, and protection from refoulement. All these objects gather support from the bill's commitment to uphold international human rights.[27]

Whatever has been captured so far is the response of the parliament towards the concerns and plight of the refugees in general. If the position specifically on Afghan refugees is to be taken into account, the Indian Parliament has, time and again, acknowledged that, on the very basis of the traditional policy of the "kindness strategy", all Afghan Nationals coming to India who are in possession of valid passports and other documents are allowed to stay in India unhindered and, subject to the fulfilment of certain conditions, are entitled to obtain visa extensions every six months.[28]

### 4.3. The Indian Judiciary: Enforcing the Customary Law of Non-Refoulement

As mentioned in the preceding sections, there is no specific or municipal legislation to deal with the rights of refugees. Because of this shortcoming, the courts of the land have read the essence of refugee rights into the text of Articles 14[29] and 21 of the Constitution of India[30], which applies to foreigners equally as well. Additionally, a combined reading of Articles 51(c) and 253 of the Indian constitution vouches for the harmonious construction between the domestic laws of the land and international law.

---

20 The Protection of Refugees and Asylum Seekers Bill, No. 290 of 2015 (India), Introduced in Lok Sabha by Rabindra Kumar Jena, MP, Available at http://164.100.47.4/billstexts/lsbilltexts/asintroduced/3024ls.pdf (accessed on 10 November 2021), Section 8.

21 The Protection of Refugees and Asylum Seekers Bill, No. 290 of 2015 (India), Introduced in Lok Sabha by Rabindra Kumar Jena, MP, Available at http://164.100.47.4/billstexts/lsbilltexts/asintroduced/3024ls.pdf (accessed on 10 November 2021), Section 35(1)(h) and Section 36(1)(b).

22 The Protection of Refugees and Asylum Seekers Bill, No. 290 of 2015 (India), Introduced in Lok Sabha by Rabindra Kumar Jena, MP, Available at http://164.100.47.4/billstexts/lsbilltexts/asintroduced/3024ls.pdf (accessed on 10 November 2021), Section 35(1)(i) and Section 36(1)(c).

23 The Protection of Refugees and Asylum Seekers Bill, No. 290 of 2015 (India), Introduced in Lok Sabha by Rabindra Kumar Jena, MP, Available at http://164.100.47.4/billstexts/lsbilltexts/asintroduced/3024ls.pdf (accessed on 10 November 2021), Section 35(1)(j) and Section 36(1)(d).

24 The Protection of Refugees and Asylum Seekers Bill, No. 290 of 2015 (India), Introduced in Lok Sabha by Rabindra Kumar Jena, MP, Available at http://164.100.47.4/billstexts/lsbilltexts/asintroduced/3024ls.pdf (accessed on 10 November 2021), Section 35(1)(e) and Section 36(1)(e).

25 The Protection of Refugees and Asylum Seekers Bill, No. 290 of 2015 (India), Introduced in Lok Sabha by Rabindra Kumar Jena, MP, Available at http://164.100.47.4/billstexts/lsbilltexts/asintroduced/3024ls.pdf (accessed on 10 November 2021), preamble.

26 The Protection of Refugees and Asylum Seekers Bill, No. 290 of 2015 (India), Introduced in Lok Sabha by Rabindra Kumar Jena, MP, Available at http://164.100.47.4/billstexts/lsbilltexts/asintroduced/3024ls.pdf (accessed on 10 November 2021), Section 2(d).

27 The Protection of Refugees and Asylum Seekers Bill, No. 290 of 2015 (India), Introduced in Lok Sabha by Rabindra Kumar Jena, MP, Available at http://164.100.47.4/billstexts/lsbilltexts/asintroduced/3024ls.pdf (accessed on 10 November 2021), preamble.

28 Lok Sabha, Un-starred Question No. 5433, Answered on 29 August 2001.

29 Article 14. Equality before law—The State shall not deny to any person equality before the law or the equal protection of the laws within the territory of India.

30 Article 21 Protection of life and personal liberty—No person shall be deprived of his life or personal liberty except according to procedure established by law.

The Supreme Court of India has always been proactive and sensitive in addressing the concerns of refugees. In the cases of *Khudiram Chakma*[31] and the *NHRC* case,[32] it expressly opined, in light of Article 14 of the Universal Declaration of Human Rights (UDHR) (1948)[33], and Article 13 of the International Covenant on Civil and Political Rights (ICCPR), that it is the "duty" of the Indian state to accord protection to refugees. In the former and latter cases, the Supreme Court held that the forceful deportation or refoulement of the Chakma refugees amounted to a violation of Article 21 of the Indian constitution. The NHRC case here holds key relevance because it furthers, or bolsters, the stand of the court in the *Khudiram* case, which is that refugees are equipped with the protection of Fundamental Rights, and, more specifically, Articles 14 and 21 in the former case. The NHRC has been actively involved in the protection and promotion of the interests of refugees in India. The NHRC has made the work on refugees its top priority and it has been taking stock of the wellbeing of refugees since 1994. It has pushed local state governments to improve their living conditions in the camps, and it has passed recommendations to the Indian government to sign and ratify the 1951 Convention and the 1967 Protocol. Additionally, the NHRC has also been actively involved in recommending that the Central Government establishes or formulates a uniform law that deals with the concerns of refugees and their grant of asylum (Dutt Tiwari 2020).

The Supreme Court, furthermore, in the case of *Committee for Citizenship Rights for Chakmas v. State,*[34] recognised the rights and entitlements of refugees to ask for citizenship, and the court also insisted that the government grant citizenship to the refugees from the Chakma community.[35]

On the very specifics of the role that is played by the High Courts, in the cases of *Gurunathan v. Govt of India*[36] and *A.C Mohd. Siddique v. Govt of India*[37], the High Court of Madras expressed its "unwillingness" to allow Sri Lankan refugees to be forced to return to Sri Lanka. One can argue that, by doing so, the court upheld the customary international law of *non-refoulement.* The High Court of Manipur expressly opined, in the case of *Nandita Haskar* v. *State of Manipur*[38], that the principle of *non-refoulement* is encompassed by Article 21, which is available to refugees as well, and that, therefore, it cannot be derogated from. The impact of the *Nandita Haskar* case is that, when the principle of *non-refoulement* is read into the fundamental right that is contained in Article 21, any government order, notification, ordinance, or legislation that pushes for the forceful deportation of refugees to their country of origin, or that pushes for their refoulement, will be contrary to the Article 21 protection that is available to refugees and, therefore, on the grounds of Article 13, it will be declared null and void.

In the very specific case of Afghan refugees, it is pertinent and relevant to note that the Supreme Court, in the case of *Maiwands Trust for Afghan Freedom,*[39] categorically ruled out, or outlawed, the forceful deportation or refoulement of *Afghan* refugees. On the point of *non-refoulment*, it should also be noted that the principle of *non-refoulment* has been read into Article 21. This jurisprudence is finally evinced by the case of *Luis De Raedt v. Union of India.*[40]

[31] *Khudiram Chakma v. State of Arunachal Pradesh, (1994) Supp (1) SCC 615.*

[32] *National Human Rights Commission v. State of Arunachal Pradesh, (1996) 1 SCC 742.* Available at https://www.refworld.org/cases,IND_SC,3f4b8de54.html (accessed on 16 October 2021).

[33] Universal Declaration of Human Rights, 1948. Available at https://www.un.org/en/about-us/universal-declaration-of-human-rights (accessed on 1 April 2022).

[34] WP (Civil) No. 510 of 2007, http://www.indiaenvironmentportal.org.in/files/Chakma%20and%20Hajong%20Tribals%20Arunachal.pdf (accessed on 23 October 2021).

[35] WP (Civil) No. 510 of 2007, http://www.indiaenvironmentportal.org.in/files/Chakma%20and%20Hajong%20Tribals%20Arunachal.pdf (accessed on 23 October 2021) at Para 16.

[36] W.P. No. 6708 and 7916 of 1992.

[37] Mr. Gurcharan Singh Sahney and others. v. Mr.Harpreet Singh Chabbra and others. (1998) 47 DRJ 74 (DB). Available at https://indiankanoon.org/doc/95458039/ (accessed on 25 October 2021).

[38] W.P. No. 6 of 2021.

[39] *Mailwand's Trust of Afghan Human Freedom v.* State of Punjab, WP (CRL) No 125 and 126 of 1986.

[40] (1991) 3 SCC 544.

The Supreme Court, as well as the High Courts, are at the forefront of upholding the customary law of *non-refoulment,* even when the Indian government has, time and again, argued that it is under no obligation to grant asylum to refugees, as it is not a signatory to the 1961 Refugee Convention, nor to the 1967 Protocol, and it has denied the binding customary nature of *non-refoulment* by relying on the national security exception (UNHCR Note on the Principle of Non-Refoulement 1997).

Lastly, the authors submit that even a cursory glance at the preceding lines makes one conclude that the High Courts and the Supreme Court have always acted in conformity with the principle of *non-refoulment,* and have enforced it, even if the legislature or the executive wings of the government have not paid it heed.

## 5. Conclusions

In this article, we have highlighted the differing courses of action that have been adopted by the executive, the legislature, and the judiciary in dealing with the rights and concerns of the refugees in India. On the basis of these trends, if the specific case of the Afghan refugees is considered, the authors submit that, unless and until any of the three pending refugee and Asylum Bill of 2015 are passed by both houses of the parliament and are made laws, and unless and until there is a paradigm shift in the material change in the standard operating procedure that has been adopted by the executive, the expected treatment of the Afghan refugees will be on a case-by-case basis for the granting of humanitarian-motivated e-visas.

Additionally, upon the expiration of six months, or of the humanitarian-motivated e-visa, as per the traditional policy of "friendship" that is acknowledged by the Indian parliament, the refugees will be deported. However, to claim the status of refugee and obtain an LTV from the executive, the application is forwarded to the MHA, which will only grant the LTV by following the standard operating procedure that is circulated by the MHA itself. As per this procedure, numerous documents must be verified in order to ascertain the forced migrant's identity so that asylum is not granted to a forced migrant with criminal antecedents. Moreover, in the *Nandita Haskar case*[41], the Supreme Court elaborated on the role of the UNHCR in conferring refugee status to asylum seekers from Afghanistan and Myanmar; accordingly, the UNHCR, in consultation with the government, can declare an Afghan forced migrant as a refugee, and can provide financial and legal assistance so that the individual can obtain an LTV from the government and can take up either employment or join an educational institution.

Even if India accords the status of refugees to the Afghan forced migrants, as it is not a signatory to the 1951 and 1967 instruments, and as it relies on the exception of national security to the customary law doctrine of *non-refoulement*, nothing is stopping the Indian government from pushing for forced deportation and for the refoulement of Afghan refugees; however, this is where the courts of the land should (and have, in earlier cases) step up. Additionally, India voted to back the GCR, and being a member of the Executive Committee of the UNHCR, it has the moral obligation to accommodate the concerns of the refugees; however, as pointed out by Professor Srinivas Burra, " . . . the GCR is non-binding and ambiguous in its nature and obligations. In the Indian context, access to protection remains an important task in the absence of clear legal and policy frameworks" (Field and Burra 2020). Moreover, the effectiveness of the GCR can only be gauged on the basis of its incorporation into the municipal laws of the state. In sum, this leads us back to the argument that the UNHCR, through the identification or registration of Afghan refugees in India, is instrumental in ensuring that India's kindness strategy is not applied in a preferential manner that favours nationals who are taking refuge from one

---

41   Nandita Haskar v. State of Manipur. Available at https://www.livelaw.in/pdf_upload/myanmarese-citizens-maniour-hc-392792.pdf (accessed on 23 December 2021). Para 17; Douglas McDonald-Norman, DEPRIVED OF LIFE:Rohingya asylum seekers and the limits of constitutional protections in India (2021), Indian Law Review.

country over those who are taking refuge from another. However, this has to be read with caution, as the mandate of the UNHCR vis-à-vis India is limited.

In this sense, the UNHCR, alongside the Indian High Courts and Supreme Court, are filling the legislative lacunae concerning registration, which includes obtaining LTVs for refugees who are applying for refugee status in India. Whether the courts or the UNHCR will prompt the Indian legislature to adopt a statute on refugee registration, the asylum application procedure, and adequate protection, is still an open question.

**Author Contributions:** Conceptualization, A.A. and N.S.; methodology, N.S. and A.A.; validation, A.A.; formal analysis, A.A. and N.S.; investigation, A.A. and N.S.; data curation, A.A.; writing—original draft preparation: N.S.; writing—review and editing: A.A. and N.S.; visualization, N.S.; supervision, A.A. All authors have read and agreed to the published version of the manuscript.

**Funding:** This research received no external funding.

**Institutional Review Board Statement:** Not applicable.

**Informed Consent Statement:** Not applicable.

**Data Availability Statement:** Not applicable.

**Conflicts of Interest:** The authors declare no conflict of interest.

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
