# Peer review of "India and Refugee Law: Gauging India’s Position on Afghan Refugees"

_laws, 2022_

Round 1

Reviewer 1 Report

The research question is how to predict if Afghan refugees would be protected by the three branches of the Indian government given that India has not ratified the GVA refugee convention. A periodizing approach is taken with tentative comparative elements. Less clear is whether the focus is on the 3 branches of government and the difference to how potential refugees will be protected or whether the focus is on the difference between non-refoulement and other forms of international protection, like the visa (e-visa and LTV). 

The article is highly original and very timely. The plight of Afghan refugees has been dealt with by UNHCR reports, but has not yet been analysed in-depth by legal scholars. In addition, there is little literature from the Global South on the GCM/GCR’s impact on national (refugee status) legislation. This piece on India covers a research gap. The material is up-to-date and comprehensively covers the questions raised. I would add Upendra Baxi’s input and the report by UNHCR https://www.unhcr.org/en-in/publications/books/5e3174c54/global-compact-on-refugees-indian-perspectives-and-experiences.html.

The article would benefit from a slightly more profound engagement with the GCM/GCR literature and policy reports, in particular as they discuss the situation for Afghan refugees in Asia and India in particular. If you could engage with Indian legal scholars discussing non-refoulement more deeply, so as to attach their line of thinking to your analysis of India’s recent case law and to flesh out more critically the challenges facing Afghan refugee protection—I suggested Upendar Baxi, but there are certainly others, including Jaja Ramji-Nogales for example.

Reviewer 2 Report

General comments: This piece takes an intriguing and less explored angle of India’s refugee law and policy, also embedding this in the context of the Global Compact on Refugees (GCR), using the situation of Afghan refugees in India as a case study. The manuscript thus combines pertinent and topical issues worthy of academic inquiry and offers a mix of macro-, meso- and micro-level analyses in an effort to shed light onto the intersections of the above-mentioned issues and topics, with a view to presenting a scientifically well-founded snapshot of the current state of affairs and salient legal features in India. The author has successfully coped with this challenging task, but some refinements in terms of structure and argumentation would be needed.

More detailed comments (following the structure of the manuscript):

  • The abstract is well-framed and highlights the aim, the main points and the key lines of reasoning of the piece. Still, it should be shortened and made a bit more accessible to uninformed readers, too.
  • The list of keywords is now too long – this needs to be reduced to 4-5 keywords.
  • As for the overall structure, the titles/sub-titles and their respective level(s) should be reconsidered – to make sure that they follow a simple logic without any hiccups and can easily guide the reader through the text, while being aligned with the underlying train of thoughts and argumentative lines. E.g. the title of sub-section 3.1 could be used as the title of section 3 (i.e. merge the two), without any further lower level of structural unit therein; section 4 should be introduced by a few lines long chapeau – to ensure better flow and strengthen context; as well as the title of section 4.1.1 should be more informative and catchy for the eye – and the same applies to the title of sub-section 4.2. In the latter unit, it is suggested to move the first two paras under section 3, as these are not connected to sub-section 4.2 [The Indian legislature] but are rather tied with the overall situation of refugees in India.
  • In terms of terminology, the author might want to be more careful in using the terms ‘refugees’ and ‘immigrants’ interchangeably or in the same context, in view of avoiding confusion; and is encouraged to explore the use of synonyms/alternative terms for ‘refugees’ (e.g. people seeking protection, persons fleeing their country of origin etc.), especially in lieu of ‘immigrants’.
  • A thorough English language proof-reading and editing is needed (there are a number of typos and grammatical errors), also to iron out too bulky sentences and phrases which are a bit cumbersome to read.
  • Section 1 [Introduction]: make sure that concrete dates and events are explicitly indicated, to help the uninformed reader and to stand the test of time (e.g. date of the latest Taliban takeover in Afghanistan; the Soviet invasion of Afghanistan etc.).
  • Section 2: some inconsistencies with Ammanullah Khan as the monarch and then Khan overthrowing the monarch in 1973 need to be sorted out. Also, please double check the date of the withdrawal of Soviet troops from Afghanistan; and explain briefly what the Taliban is at the first mention. Add references & concrete figures to the statement concerning the substantial number of Afghans fleeing their country in 2021.
  • Section 3: spell out ‘LTV’ at the first occasion where it is mentioned.
  • Section 4: the standard operating procedures adopted by the MHA should be referenced in a footnote. The same need for referencing applies to the mandate of the UNHCR re. exclusion of some nationalities. Make sure that ‘UNHCR’ is correctly spelled across the whole text. In sub-section 4.1.1., the statement of ‘no humanitarian organisation operates on the principle of ‘neutrality’’ seems to be too strong – see e.g. the ICRC of which this is a core guiding principle and is not contested by stakeholders. In the same part, please add references/sources to India’s membership in ExCom of UNHCR. In sub-section 4.2, could you please explain to what extent the quoted provisions of Indian domestic laws governing the status of foreigners apply to refugees? Also, what is the relationship between the three pending bills relating to the status/situation of refugees; how do they relate to each other? Some more elaboration would be welcome here, along with some information on the legislative process that could lead to their adoption. At the end of each sub-section discussing the position of the Indian executive, legislature and judiciary, some critical evaluation and flashing the possible way forward would be helpful.
  • Overall, more engagement with the Global Compact on Refugees here and there could further embed this piece into the overall theme of the Special Issue and could strengthen its connections with the GCR.
